# Critical Role of 3′-Downstream Region of *pmrB* in Polymyxin Resistance in *Escherichia coli* BL21(DE3)

**DOI:** 10.3390/microorganisms9030655

**Published:** 2021-03-22

**Authors:** Fuzhou Xu, Atsushi Hinenoya, Ximin Zeng, Xing-Ping Li, Ziqiang Guan, Jun Lin

**Affiliations:** 1Department of Animal Science, University of Tennessee, Knoxville, TN 37996, USA; xufuzhou@iasbaafs.net.cn (F.X.); hinenoya@vet.osakafu-u.ac.jp (A.H.); xzeng3@utk.edu (X.Z.); lxp@haust.edu.cn (X.-P.L.); 2Institute of Animal Husbandry and Veterinary Medicine, Beijing Academy of Agriculture and Forestry Sciences, Beijing 100097, China; 3Graduate School of Life and Environmental Sciences & Asian Health Science Institute, Osaka Prefecture University, Osaka 599-8531, Japan; 4College of Animal Science and Technology, Henan University of Science and Technology, Luoyang 471000, China; 5Department of Biochemistry, Duke University Medical Center, Durham, NC 27708, USA; ziqiang.guan@duke.edu

**Keywords:** polymyxin resistance, two-component regulatory system, mRNA decay, lipid A modification

## Abstract

Polymyxins, such as colistin and polymyxin B, are the drugs used as a last resort to treat multidrug-resistant Gram-negative bacterial infections in humans. Increasing colistin resistance has posed a serious threat to human health, warranting in-depth mechanistic research. In this study, using a functional cloning approach, we examined the molecular basis of colistin resistance in *Escherichia coli* BL21(DE3). Five transformants with inserts ranging from 3.8 to 10.7 kb displayed significantly increased colistin resistance, three of which containing *pmrB* locus and two containing *pmrD* locus. Stepwise subcloning indicated that both the *pmrB* with a single G361A mutation and at least a 103 bp downstream region of *pmrB* are essential for conferring colistin resistance. Analysis of the mRNA level and stability showed that the length of the downstream region drastically affected the *pmrB* mRNA level but not its half-life. Lipid A analysis, by mass spectrometry, revealed that the constructs containing *pmrB* with a longer downstream region (103 or 126 bp) have charge-altering l-4-aminoarabinose (Ara4N) and phosphoethanolamine (pEtN) modifications in lipid A, which were not observed in both vector control and the construct containing *pmrB* with an 86 bp downstream region. Together, the findings from this study indicate that the 3′-downstream region of *pmrB* is critical for the PmrB-mediated lipid A modifications and colistin resistance in *E. coli* BL21(DE3), suggesting a novel regulatory mechanism of PmrB-mediated colistin resistance in *E. coli*.

## 1. Introduction

Polymyxins, such as colistin (also known as “polymyxin E”) and polymyxin B, are polycationic peptide antibiotics with broad-spectrum activity against Gram-negative bacterial pathogens (such as *Escherichia coli*, *Pseudomonas aeruginosa*, *Acinetobacter baumannii* and *Klebsiella pneumoniae*). Colistin was introduced into clinical usage in the 1960s but was replaced by other antibiotics due to concerns of nephrotoxicity and neurotoxicity [1]. Since the 1990s, with the advent of multidrug-resistant (MDR) Gram-negative bacteria, colistin has been re-introduced and considered as one of the drugs used as a last resort to treat MDR Gram-negative bacterial infections in humans [2]. However, colistin resistance has been arising in various significant Gram-negative bacteria, posing a serious threat to the clinical treatment of MDR pathogens [3].

Colistin acts on Gram-negative bacteria by disrupting the integrity of the outer membrane of Gram-negative bacteria. Cationic colistin can bind negatively charged lipopolysaccharide (LPS), which is anchored at the outer leaflet of the outer membrane through lipid A moiety. In normal conditions, divalent cations, such as Mg^2+^ and Ca^2+^, form ionic bridges between nearby lipid A to stabilize the outer membrane [4]. Cationic colistin can competitively displace Mg^2+^ and Ca^2+^ from the lipid A, because colistin displays several orders of magnitude higher affinities to lipid A than those divalent cation ions [5]. The displacement of divalent cation ions by colistin destabilizes the LPS molecules, therefore weakening the permeability barrier function of the outer membrane [1,6].

To date, several strategies used by bacteria to resist colistin have been observed and investigated [7,8]. A major and conserved resistance mechanism is LPS modification, such as modifications of lipid A with cationic l-4-aminoarabinose (Ara4N) and phosphoethanolamine (pEtN) via the two-component regulatory system PhoP-PhoQ (PhoPQ) and PmrA-PmrB (PmrAB) [8]. The modification of lipid A with specific cationic moiety (Ara4N or pEtN) decreases the negative charge of the lipid A, consequently leading to resistance to the cationic polymyxin antibiotics [8]. Specifically, the PmrAB system is considered as a major regulatory system in polymyxin resistance [9,10,11]. PmrB, a sensor kinase, responds to different signals, such as cationic antimicrobial peptides, mildly acidic pH and high Fe^3+^ concentration [10,12,13], and then promotes phosphorylation of the PmrA regulator. PmrD interacts with phosphorylated PmrA, protecting it from dephosphorylation [14]. Phosphorylated PmrA transcriptionally activates downstream effectors, such as two key enzymes, EptA (pEtN modification enzyme) and ArnT (Ara4N modification enzyme), which catalyze lipid A modification [9,15]. The PhoPQ system responds to the signals of low Mg^2+^ concentration and cationic antimicrobial peptides [16,17,18] and can indirectly affect PmrAB-activated genes via the PhoP-activated connector protein PmrD [19,20].

*Escherichia coli* BL21 (DE3) (designated as “BL21” hereinafter) is a model organism for research as well as a workhorse for biotechnology, such as the production of recombinant proteins. During our recent colistin research [21], we were surprised to observe that BL21 displayed clinical resistance to colistin (MIC = 16 µg/mL), which was confirmed in two BL21 strains purchased from different suppliers, Stratagene (La Jolla, CA, USA) and Novagen (Madison, WI, USA). In this study, the colistin resistance mechanism of BL21 was further investigated using a functional cloning approach in conjunction with genetic manipulations and lipid A species analysis. Our findings indicate that the 3′-downsstream region of *pmrB* plays a critical role in the expression and functionality of *pmrB* in BL21.

## 2. Materials and Methods

### 2.1. Bacterial Strains, Plasmids and Culture Conditions

The major bacterial strains and plasmids used in this study are listed in Table 1. *E. coli* strains were grown in Luria-Bertani (LB) broth or Mueller Hinton (MH) broth (Becton, Dickinson and Company, Franklin Lakes, NJ, USA) with shaking (250 rpm) or on agar at 37 or 32 °C overnight. When needed, culture media were supplemented with ampicillin (100 μg/mL, Fisher Scientific, Pittsburgh, PA, USA), kanamycin (30 μg/mL, Fisher Scientific, Pittsburgh, PA, USA) and/or colistin sulfate (2 μg/mL, ACROS, Geel, Belgium).

### 2.2. Transformation of TOP10 with Putative Colistin-Resistant Genes and Different Lengths of pmrB from BL21

Genes of *eptA*, *eptB*, *cptA*, *opgE*, *pmrD*, *pmrAB* [6,25] and different lengths of *pmrB*, were cloned into the cloning vector pZE21 [22] and introduced to colistin-susceptible host strain TOP10. Briefly, the selected genes were PCR amplified with corresponding primers (listed in Table 2) and PfuUltra DNA polymerase (Stratagene) using genomic DNA of BL21 as the template. The blunt-ended PCR products were digested with *Bam*HI and ligated into the *BamH*I/*EcoR*V double digested pZE21. Different lengths of *pmrB* PCR products were digested with corresponding restriction enzymes (as indicated in Table 2) and ligated into the same enzyme digested pZE21. These ligation mixes were transformed into *E. coli* TOP10. Plasmids were extracted from transformants and validated by Sanger sequencing with primer pZE-F and pZE-R (Table 2). Those transformants were streaked onto the LB plates containing 50 μg/mL of kanamycin and 2 μg/mL of colistin to test colistin susceptibility.

### 2.3. Functional Cloning of Colistin-Resistant Elements from BL21

The colistin-resistant BL21 strain was used as the genomic DNA donor for functional cloning as previously described [26]. Total bacterial genomic DNA of BL21 was extracted using the Wizard^®^ Genomic DNA Purification Kit (Promega, Madison, WI, USA). Approximately 20 µg of bacterial genomic DNA was sheared using a Covaris M220 Focused ultrasonicator (Covaris Inc., Woburn, MA, USA). The fragment size was determined on 0.8% agarose gels (with 0.4 µg/mL ethidium bromide). The gel slices with sizes ranging from 1 to 4 kb were excised, and the DNA was extracted using QIAquick^®^ Gel extraction kit (Qiagen, Valencia, CA, USA). Extracted fragments were then blunt-ended using the End-it^®^ end repair kit (Epicentre, Middleton, WI, USA). End-repaired fragments were purified from the reaction mix by using QIAquick PCR purification kit (Qiagen) and ligated to an *Eco*RV-digested, dephosphorylated pZE21-MCS cloning vector [22] using the Fast-Link^®^ ligation kit (Lucigen, Middleton, WI, USA) in a reaction mix containing 4 μL of vector DNA (200 ng), 12 μL of sheared DNA (600 ng), 2 μL of 10× Fast-Link buffer, 1 μL of 10 mM ATP and 1 μL of T4 ligase (2 U/µL). The ligation mix was incubated overnight at room temperature (22 °C). A total 2 μL of the ligation mix was electroporated into 50 μL of ONE-SHOT^®^ TOP10 electrocompetent cells (Invitrogen). Electroporation was conducted using the MicroPulser Electroporation system (Bio-Rad, Hercules, CA, USA) with the pre-programmed setting Ec1 (for *E. coli* in 0.1 cm cuvette, 1.8 kV, 1 pulse). Cells were immediately recovered in 1 mL of SOC medium by shaking at 250 rpm for 1 h at 37 °C. After recovery, 10 μL of the recovered cells was used for determining the library size. Cells were serially diluted in LB broth and plated on LB agar plates containing 50 μg/mL of kanamycin for determination of total colony-forming units (CFUs) of the 1 mL recovered cells. The size of the library was then estimated by multiplying the total CFU of recovered cells and the average size (3 kb) of the inserted fragments. The rest of the transformation mixes were inoculated into 10 mL LB broth supplemented with 50 μg/mL of kanamycin and grown overnight at 37 °C. The overnight cultures were spread onto the LB agar plates (100 μL/each) containing 50 μg/mL of kanamycin and 2 μg/mL of colistin for screening. Colonies was streaked again onto LB agar plates containing 50 μg/mL of kanamycin and 2 μg/mL of colistin to confirm the colistin resistance. Real positive colonies were grown in LB broth containing 50 μg/mL of kanamycin and subsequently subjected to plasmid extraction. Plasmids were sequenced using Sanger sequencing.

### 2.4. Site-Directed Mutagenesis of the Genes in BL21

Lambda-red-based homologous recombination technology using a pSIM6 vector (supplied by Dr. Donald Court) [24] was used to knockout target genes in BL21. Mutational fragments encompassing an FRT-*kan*-FRT cassette in pKD13 template plasmid [23] with 50 nt homologous arms immediately flanking each targeted region (primers in Table 2) were electroporated into 50 µL of heat-shock-induced electrocompetent cells using MicroPulser Electroporation Apparatus (Bio-Rad) and a 0.1 cm gapped electroporation cuvette (Bio-Rad) with the EC1 program (1.8 kV). Recombinants were selected for kanamycin (*kan*) resistance (30 µg/mL) at 32 °C for 1–2 days, and then streaked onto LB plates and incubated at elevated temperature (37 °C) to remove pSIM6. The mutation was further verified by PCR using flanking primers (Table 2) and k1 [23].

### 2.5. Antimicrobial Susceptibility Test

The susceptibilities of *E. coli* strains to colistin sulfate were determined by a standard microtiter broth dilution method with an inoculum of 10^6^ CFU/mL as previously described [27,28]. Minimum inhibitory concentration (MIC) for colistin was determined by the lowest concentration of the antimicrobial showing complete inhibition of bacterial growth after 18 h incubation at 37 °C.

### 2.6. Quantitative Real-Time RT-PCR (qRT-PCR) Analysis of pmrB mRNA Level at Steady Phase

The TOP10 derivative strains JL1088, JL1381, JL1507 and JL1509 were used to determine the *pmrB* mRNA level at steady phase. These four strains contain parent pZE21 vector, pPmrB, pUS150-PmrB-DS86 and pUS150-PmrB-DS126 (Figure 1), respectively. The primer pair, RT-PmrB-BL21SNP-F3 and RT-PmrB-BL21SNP-R3 (Table 2), which was designed by targeting the G361A mutation region in *pmrB*_BL21_ using the method as previously described [29], can specifically detect the *pmrB* from BL21 while barely amplifying the *pmrB* in TOP10 chromosome. To prepare RNA, *E. coli* was cultured to logarithm phase (OD_600nm_ = 0.5–1.0), and 500 µL of the *E. coli* culture was taken and mixed with 1 mL RNAprotect bacterial reagent (Qiagen). Total RNA from each sample was extracted using an RNeasy mini kit (Qiagen). Genomic DNA was removed with DNase I (Qiagen) digestion, and RNA was re-purified using an RNeasy mini kit (Qiagen). Subsequent qRT-PCR analysis was carried out as previously described [28,30]. 16S rRNA gene was targeted as an internal control using primers 16S-F and 16S-R (Table 2). The relative changes (expressed as −ΔΔCt) of *pmrB* mRNA levels between JL1088 and other strains were calculated as described before [28]. Each sample was measured in triplicate, and three independent experiments were performed.

### 2.7. Determination of Decay Rate of pmrB mRNA

TOP10 derivative strains JL1507 and JL1509, which contain pUS150-PmrB-DS86 and pUS150-PmrB-DS126, respectively, were used to determine the *pmrB* mRNA decay rate (half-life) using the rifampin arrest method as previously described [30] with some modifications. Briefly, *E. coli* overnight cultures were inoculated into fresh medium containing kanamycin (30 μg/mL), and grew to logarithm phase (OD_600nm_ = 0.5–1.0) in an orbital shaker (250 rpm) at 37°C. Rifampin (500 μg/mL) was then added to the cultures and mixed well. At different time points (0, 1, 2, 5, 10, 20, 40 and 70 min) after the addition of rifampin, 500 µL of the *E. coli* culture was taken and added to 1 mL RNAprotect bacterial reagent (Qiagen). Total RNA from each sample was extracted, digested with DNase I and re-purified as described above. The primer pair, RT-PmrB-BL21SNP-F3/RT-PmrB-BL21SNP-R3 and 16S-F/16S-R (Table 2), was used in qRT-PCR to determine the mRNA abundance as described previously [31]. Each sample was measured in triplicate, and three independent experiments were performed.

### 2.8. Lipid A Profile Analysis

Lipid A was extracted from bacterial pellets using the Bligh–Dyer method [32,33]. Approximately 200 mL of logarithmic phase (OD_600nm_ = 0.8–1) cell culture of each strain was used for lipid A extraction. Once extracted, lipid A samples were subjected to liquid chromatography/electrospray ionization-mass spectrometry (LC/ESI-MS) analysis at Duke Medical Center as previously described [34,35]. Briefly, normal phase LC-ESI MS of the lipid extracts was performed using an Agilent 1200 Quaternary LC system coupled to a high resolution TripleTOF5600 mass spectrometer (Sciex, Framingham, MA, USA). Chromatographic separation was performed on a Unison UK-Amino column (3 μm, 25 cm × 2 mm) (Imtakt USA, Portland, OR, USA). Lipids were eluted with mobile phase A, consisting of chloroform/methanol/aqueous ammonium hydroxide (800:195:5, *v/v/v*); mobile phase B, consisting of chloroform/methanol/water/aqueous ammonium hydroxide (600:340:50:5, *v/v/v/v*); and mobile phase C, consisting of chloroform/methanol/water/aqueous ammonium hydroxide (450:450:95:5, *v/v/v/v*), over a 40 min run, performed as follows: 100% mobile phase A was held isocratically for 2 min and then linearly increased to 100% mobile phase B over 14 min and held at 100% B for 11 min. The mobile phase composition was then changed to 100% mobile phase Cover 3 min and held at 100% C for 3 min, and finally returned to 100% A over 0.5 min and held at 100% A for 5 min. The LC eluent (with a total flow rate of 300 µL/min) was introduced into the ESI source of the high resolution TF5600 mass spectrometer. MS and MS/MS were performed in negative ion mode, with the full-scan spectra being collected in the *m/z* 300-2000 range. The MS settings are as follows: ion spray voltage (IS) = −4500 V, curtain gas (CUR) = 20 psi, ion source gas 1 (GS1) = 20 psi, de-clustering potential (DP) = −55 V, and focusing potential (FP) = −150 V. Nitrogen was used as the collision gas for tandem mass spectrometry (MS/MS) experiments. Data analysis was performed using Analyst TF1.5 software (Sciex, Framingham, MA, USA).

### 2.9. Statistical Analysis

Real-Time PCR results for analysis of the *pmrB* mRNA level at steady phase were tested by analysis of variance (ANOVA) using SAS software (v. 9.03). Two-way ANOVA followed by a least significant difference (LSD) test was used to assess the significance of differences among relative changes (expressed as −ΔΔCt) of *pmrB* mRNA levels. The qRT-PCR data used to calculate the *pmrB* mRNA half-life were analyzed using GraphPad Prism software (GraphPad, San Diego, CA, USA). The mRNA half-life was determined by one-phase decay using a nonlinear regression model, and the Kolmogorov–Smirnov statistic was used to determine if differences in distributions were significant, as previously described [30]. Levels of significance for *p* value are 5% (0.05).

## 3. Results

### 3.1. The Genes from BL21 Implicated in Polymyxin Resistance Failed to Confer Colistin Resistance in TOP10

Initially, we examined a panel of genes potentially involved in polymyxin resistance by using the same functional rescuing approach previously described [21]. The open reading frames (ORFs) of selected genes (*eptA*, *eptB*, *cptA*, *opgE*, *pmrD* and *pmrA*-*pmrB*) were cloned from BL21 into a pZE21 expression vector, respectively; the expression of the cloned gene was driven by the strong promoter PLtetO-1 [22]. None of these genes could lead to increased MIC of colistin in the TOP 10 strain (data not shown). In particular, based on the examination of the BL21 genome sequence (GenBank Accession #: AM946981.2), we observed a single nucleotide mutation in *pmrB* gene (G361A) that led to Glu-121-Lys aa substitution in PmrB. However, the plasmid bearing *pmrB*_BL21_ failed to confer colistin resistance in TOP10. This preliminary study led us speculate that a new polymyxin resistance mechanism may exist in BL21 and prompted us to systematically identify relevant genetic loci using a functional cloning approach, which was successfully used in our previous study [26].

### 3.2. Discovery of Colistin-Resistant Determinants Using Functional Cloning

A total of five transformants (JL1365-1369; Table 1 and Table 3) were identified and confirmed to grow on LB agar plates supplemented with 2 µg/mL of colistin. Plasmid from each transformant was extracted, and the insert region was sequenced. The features of the inserted region are summarized in Table 3. Of note, the majority of the constructs (except JL1366) contained chimeric segments from distant regions in BL21, which was due to the ligation issue (Table 3). Gene annotation revealed a panel of genes that are known to involve polymyxin resistance (red text in Table 3). For example, two major loci, *pmrD* and *pmrB*, were commonly observed in these transformants. The *pmrD* and its adjacent regions (800 bp upstream and downstream of *pmrD*) were identical between BL21 and *E. coli* K12 (MG1655), a colistin susceptible strain. This evidence together with the inability of BL21-derived PmrD to confer colistin resistance in TOP10 suggests that other distant genetic elements (e.g., *arnF*) in those plasmids may contribute to colistin resistance.

It was intriguing that the *pmrB*_BL21_ gene that was ruled out in our initial preliminary screening was shown in three resistant constructs (JL1366, JL1367 and JL1369) (Table 3). Comparison of the sequenced *pmrB*_BL21_ in these constructs to those from colistin susceptible *E. coli* K12 strains (e.g., MG1655, TOP10 and DH5α) further confirmed the existence of a single G361A mutation in the ORF of *pmrB*_BL21_. This finding suggests that PmrB is indeed a key player for the acquired colistin resistance in BL21, and prompted us to continue to examine the role of *pmrB*_BL21_ gene and its 3′-downstream region in colistin resistance in BL21.

### 3.3. PmrB but Not PhoQ Is Required for Colistin Resistance in BL21

To accurately define the critical role of *pmrB* in the colistin resistance of BL21, different approaches were used to create isogenic *pmrB* mutant of BL21. We first used pKD46 as a lambda red recombinase provider [23], which was induced by adding L-arabinose and has been reported to work successfully in *E. coli* K-12 strains. However, we did not obtain desired mutants, although we tested multiple conditions (concentration and timing of L-arabinose inducer, and different length of homologous arms). Given that BL21 is *araBAD*^+^ and can metabolize l-arabinose [36], lambda red recombinase may not be efficiently induced by the supplementation of L-arabinose to generate the desired mutant. To address this technical challenge uniquely associated with BL21, pSIM6 plasmid (a generous gift from Dr. Donald L. Court at National Cancer Institute), which produces lambda red recombinase upon heat induction [24], was subsequently used in this study. Using pSIM6 plasmid, we obtained isogenic *pmrB* mutant successfully. As shown in Table 4, the inactivation of *pmrB* in BL21 drastically reduced colistin MIC to 0.5 μg/mL when compared to parent BL21 (colistin MIC = 16 μg/mL). Since a PhoPQ two-component regulatory system can indirectly activate PmrAB-regulated genes via the connector protein PmrD [9,10,37], an isogenic *phoQ* mutant of BL21 (JL1435; Table 1) was also generated using the same mutagenesis strategy. As shown in Table 4, the *phoQ* mutant still displayed colistin resistance at a level comparable to that of its parent BL21 (MIC = 16 μg/mL). Thus, the PmrB rather than PhoQ plays a critical role in colistin resistance in BL21.

### 3.4. Identification of Critical 3′-Downstream Region of pmrB Required for Colistin Resistance

The insert in the plasmid carried by JL1369 contains a 2369 bp fragment (annotated as *purT*-*eda*-*edd*) and a 2900 bp fragment (annotated as *proP*-*pmrB*-*pmrA*) (Table 3). Since the abovementioned site-directed mutagenesis study demonstrated that *pmrB* is critical for colistin resistance, we then performed stepwise subcloning focusing on *pmrB* locus. The 2897 bp fragment (Figure 1A), which contains partial *proP*, full length *pmrB* and partial *pmrA*, was PCR amplified and cloned into pZE21; the resulting recombinant plasmid pPmrA-PmrB-ProP (Table 1) conferred colistin resistance in *E. coli* TOP10 (MIC = 16 µg/mL). Further subcloning produced a much smaller fragment that could still confer the same level of colistin resistance in TOP10 (JL1431, MIC = 16 µg/mL), which includes 150 bp upstream and 176 bp downstream regions of *pmrB* (Figure 1A and Table 4). Notably, the corresponding fragment derived from *E. coli* MG1655, which does not have the G361A single nucleotide mutation observed in BL21, failed to confer colistin resistance in TOP10 (Table 4). This finding demonstrated that the G361A single nucleotide mutation in BL21 caused a functionality change in PmrB, leading to acquired colistin resistance in BL21.

Given that the TOP10 derivatives, JL1371 and JL1381, which carry *pmrB-pmrA* ORF and *pmrB* ORF, respectively, displayed the same susceptibility to colistin (MIC = 0.5 µg/mL), the 176 bp downstream sequence of *pmrB* clearly played a critical role in PmrB-mediated colistin resistance in BL21. It was reported that the 3′-untranslated region (UTR) of a specific gene may form secondary structures, consequently affecting the expression and function of the preceding genes [30,38]. Analysis of 3′-UTR of *pmrB* using RNAfold WebServer [39] revealed multiple stem-loop structures (Figure 1B). Therefore, multiple sites, which represent transition points between stems and loops within the 3′-downstream +176 bp region (such as +34, +86, +103, +126, +134 bp; Figure 1A,B), were chosen for further subcloning, leading to a panel of constructs (Figure 1A). As shown in Table 4, up to 34 or 86 bp of the *pmrB* downstream region still failed to confer colistin resistance in TOP10. However, up to 103 bp of the downstream region of *pmrB* conferred a significantly increased colistin resistance (MIC = 4 µg/mL), while the longer regions (+126 or +134 region) could confer colistin resistance to the same level as that observed for the +176 bp region (MIC = 16 µg/mL; Table 4). These results clearly indicate that 3′-UTR of *pmrB* plays an important role in colistin resistance in *E. coli*.

### 3.5. 3′-Downstream Region of pmrB Modulates the Expression Level of pmrB

The findings from the abovementioned stepwise subcloning showed that the function of *pmrB*_BL21_ was affected by the length of the 3′-downstream region. It is likely that this phenotypic response was caused by the difference in the expression levels of *pmrB* in different constructs. To test this, the mRNA level of *pmrB* was assessed in selected representative strains using qRT-PCR with a pair of PCR primers (RT-PmrB-BL21SNP-F3 and RT-PmrB-BL21SNP-R3; Table 2) that specifically target the *pmrB*_BL21_ containing a single G361A mutation. Initial RT-PCR using TOP10 and BL21 genomic DNA as templates demonstrated that this pair of specific primers could efficiently distinguish the *pmrB*_BL21_ and the TOP10 chromosomal *pmrB* with only a single nucleotide difference. Specifically, using the primer pair together with a similar amount of genomic DNA (4 ng/µL) as a template, the Ct value for the amplicon from BL21 was 15.1, while the Ct value for the amplicon from TOP10 was 32.0. In addition, endogenous *pmrB* in JL1088 the control strain (Table 1; TOP10/pZE21) was also barely detected by this pair of primers; therefore, JL1088 was used as the baseline for the comparison of the level of *pmrB*_BL21_ in the three selected TOP10 derivative strains (Figure 2A). The JL1381 (TOP10/pPmrB), which only contains *pmrB* ORF, displayed the highest level of *pmrB*_BL21_ (−ΔΔCt = 16.5) (Figure 2A). The mRNA level of *pmrB*_BL21_ in JL1507 (−ΔΔCt = 6.5) that carries 86 bp of 3′-UTR was significantly lower (approximately 315-fold lower, *P* < 0.05) than the mRNA level of *pmrB*_BL21_ in JL1509 (−ΔΔCt = 14.8) that carries longer 3′-UTR (+126) (Figure 2A).


### 3.6. 3′-Downstream Region Did Not Affect mRNA Stability of pmrB

Since a significantly low level of *pmrB*_BL21_ mRNA was observed in JL1507 when compared to that in JL1509 (Figure 2A), the 3′-UTR length may modulate and determine the stability of *pmrB*_BL21_ mRNA, as observed in the regulation of other genes [30,38]. To test this hypothesis, the half-life of *pmrB* in JL1507 and JL1509 was further measured by rifampin arrest treatment followed by qRT-PCR. As shown in Figure 2B, after adding rifampin to stop new mRNA synthesis, the mRNA level of in both strains started to decrease rapidly at a similar rate. For JL1507 that carries plasmid pUS150-PmrB-DS86, the half-life of *pmrB* was 1.0 min (goodness-of-fit R^2^: 0.9786), while the *pmrB* mRNA half-life in JL1509 was 0.83 min (goodness-of-fit R^2^: 0.9572). The difference in mRNA in the half-life between JL1507 and JL1509 was not statistically significant (*p* > 0.05), indicating that the mRNA decay rate in the two strains is similar.

### 3.7. 3′-Downstream Region of pmrB Modulates Lipid A Modification

Lipid A profiles of selected strains carrying different lengths of the 3′-downstream region of *pmrB*_BL21_ were analyzed using LC/ESI-MS. As predicted, there were no pEtN and Ara4N modifications in the JL1088 vector control strain (Figure 3A) and JL1507 that carries 86 bp of 3′-UTR (Figure 3B), consistent with their susceptibility to colistin (Table 4). In contrast, JL1508 and JL1509, which contain 103 and 126 bp of the *pmrB* downstream region, respectively, displayed charge-altering modifications in their lipid A, including a single modification with Ara4N or pEtN, double modifications with the same moiety pEtN and double modifications with different moieties (Ara4N and pEtN) (Figure 3C,D). These positively charged moieties reduced the negative charges of lipid A and limited its interaction with colistin, which is cationic. A close comparison of the lipid A profiles of JL1508 and JL1509 showed their modification differences. Specifically, JL1508 appeared to have more Ara4N (or less pEtN) modification than JL1509, suggesting that ArnT (responsible for the Ara4N modification) is more active in JL1508, while EptA (responsible for the pEtN modification) is more active in JL1509.

## 4. Discussion

In this study, the colistin resistance mechanism of *E. coli* BL21, a bacterial strain widely used for research and commercial biotechnology, was explored using a functional cloning approach. A total of five colistin-resistant clones were identified, and two major loci, *pmrB* and *pmrD* (Table 3), were found to be potentially responsible for the colistin resistance in BL21. In this study, the role of *pmrB*_BL21_ in colistin resistance was further investigated. The findings from a panel of molecular manipulations (site-directed mutagenesis and complementation) provide compelling evidence that the G361A single nucleotide mutation in *pmrB*_BL21_ caused a functionality change in PmrB, leading to acquired colistin resistance in BL21; this is consistent with other studies showing that a single point mutation in PmrAB and/or PhoPQ could result in sustained polymyxin resistance [7,8]. However, for the first time, we also demonstrated the critical role of the 3′-downstream region of *pmrB* in colistin resistance in *E. coli* BL21 using delicate molecular manipulation in conjunction with comprehensive characterizations, such as the MIC test, the mRNA level and stability analyses, and LC/ESI-MS analysis of the lipid A profile. Up to 126 bp of *pmrB*_BL21_ 3′-UTR was needed to confer full colistin resistance (MIC = 16 μL/mL), while the *pmrB*_BL21_ with 86 bp or less 3′-UTR failed to confer any colistin resistance in susceptible TOP10 host strain (Table 4). Different lengths of 3′-UTR caused significant difference in the expression level of *pmrB*_BL21_ (Figure 2A) but did not affect mRNA stability (Figure 2B). Consistent with MIC data, as expected, the *pmrB*_BL21_ with long 3′-UTR is not only required for acquired colistin resistance but also extensive modifications of lipid A (Table 4 and Figure 3).

It is interesting that the strain bearing *pmrB*_BL21_ ORF only (JL1381) displayed an exceptionally high level of *pmrB*_BL21_ mRNA (Figure 2A) but was still susceptible to colistin (MIC = 0.5 μg/mL; Table 4). PmrB, despite its annotation as a kinase, also has phosphatase activity to remove the phosphoryl group. In particular, it has been reported that *E. coli* PmrB displays higher phosphatase activity towards phosphor-PmrA than the *Salmonella* PmrB [40]. Therefore, the excessive production of PmrB_BL21_ by pPmrB in *E. coli* TOP10 may also dephosphorylate the activated phosphor-PmrA more quickly [14], leading to a lack of colistin resistance-related lipid A modification due to the dramatically decreased expression of the PmrA-activated lipid A modification enzymes, such as EptA and ArnT. This hypothesis will be examined in future studies.

PmrD, a PhoP-activated small protein that connects PhoPQ and PmrAB, two component regulatory systems [14], can also shift the equilibrium between the phosphorylated-PmrA and dephosphorylated-PmrA and determine the final level of phosphor-PmrA [14]. PmrD can specifically interact with phosphorylated PmrA, protecting it from PmrB-promoted dephosphorylation [14]. However, the BL21-derived PmrD itself does not seem to be a contributor to the acquired colistin resistance in two transformants (JL1365 and JL1368; Table 3) that contain *pmrD* locus. In particular, we demonstrated that the overexpression of *pmrD*_BL21_ did not confer colistin resistance in TOP10. The components other than PmrD in the two *pmrD*-carrying recombinant plasmids, such as the alternative PmrD regulatory element [41], may contribute to colistin resistance. To test this hypothesis, in the future, the same subcloning approach will be used to identify specific component(s) in the two recombinant plasmids that are required for colistin resistance.

With respect to the 3′-UTR region, we were specifically focused on the role of the secondary structure on *pmrB* expression and functionality in this study. It is important to mention that other genetic elements in the 3′-UTR region may also play a role in modulating the expression and function of *pmrB*. It was reported that one-third of mRNAs of *Staphylococcus aureus* carry 3′-UTRs longer than 100 nt [42]. In the mRNA of *icaR*, there is a base pairing between the 3′-UTR and the 5′ Shine-Dalgarno (SD) regions, which interferes with the formation of the translation initiation complex and generates a double-stranded hairpin substrate for RNase III [42]. In a different scenario, the 3′-UTR can also serve as a target of bacterial sRNAs [43] or a docking platform for RNA chaperone binding, consequently protecting mRNA from RNase II degradation [44]. Interestingly, the RNA chaperone ProQ, which binds to sRNAs and mRNA 3′-UTRs, was originally reported as an important modulator of *proP*, which is immediate downstream of *pmrB* (Figure 1A). Notably, the intergenic region between *E. coli pmrB* and *proP* is the 3′-downstream region of both genes. Further investigation is highly warranted to unveil possible roles of sRNA or RNA chaperones in PmrB-mediated polymyxin resistance in Gram-negative bacteria.

In addition to the RNA-related *cis*-elements discussed above, a small ORF (90 bp) that is transcribed in an opposite direction of *pmrB* with 25 bp overlapping region may contribute to the PmrB-mediated colistin resistance in BL21. Annotation analysis indicated that this gene encodes a small protein homologous to PmrR, which has only been reported in limited *Salmonella* studies [13,45]. In *Salmonella*, PmrR was observed to inhibit the activity of LpxT, an enzyme responsible for generating diphosphorylated lipid A at the 1-position and, therefore, increasing surface negative charge. The inhibition of LpxT by PmrR could prevent the increasing surface negative charge, leading to decreased affinity for Fe^3+^, the inducing signal of PmrAB systems. Thus, PmrR in *Salmonella* was proposed to participate in the negative feedback loop to the PmrAB pathway by reducing the binding affinity of inducing Fe^3+^ to sensor PmrB and consequently downregulating the transcription of PmrA-activated genes [13,45]. However, in this *E. coli* study, we observed that the long 3′-downstream region (such as in JL1508 and JL1509) where *pmrR* resides is required for the enhanced expression of PmrB, leading to lipid A modifications and increased resistance to colistin, which does not fit into the reciprocal control model of PmrR and PmrB [45]. We speculate that PmrR may behave differently in *E. coli* and *Salmonella* to modulate PmrAB-mediated polymyxin resistance, as we observed for PmrD [40,46]. This hypothesis will be examined in the future.

## Figures and Tables

**Figure 1 microorganisms-09-00655-f001:**
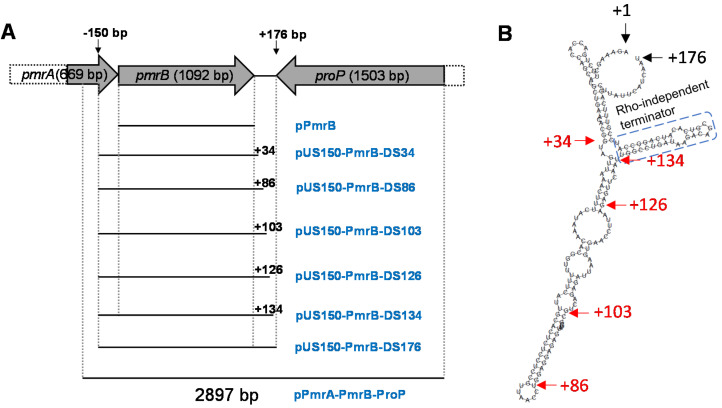
Diagram of subcloning strategy and the predicted stem-loop structure of 3′-downstream region of *pmrB*. (**A**) Stepwise subcloning and corresponding plasmid constructs (plasmid name in blue). Vertical dash lines cover the cloned major regions selected for subcloning. −150 bp, 150 bp upstream of *pmrB* ORF. The length of region downstream of *pmrB* ORF in different constructs was indicated by the number with “+” prefix. (**B**) Stem-loop structure prediction of 3′-downstream region of *pmrB*. The starting (+1) and ending (+176) of the 3′-downstream region of *pmrB* are indicated with black arrows. The transition sites between some stems and loops are indicated with arrows followed by the distance from the *pmrB* stop codon (in red); these sites were also selected for creating subcloning plasmids (detailed in A panel). The blue dash box indicates predicted Rho-independent terminator.

**Figure 2 microorganisms-09-00655-f002:**
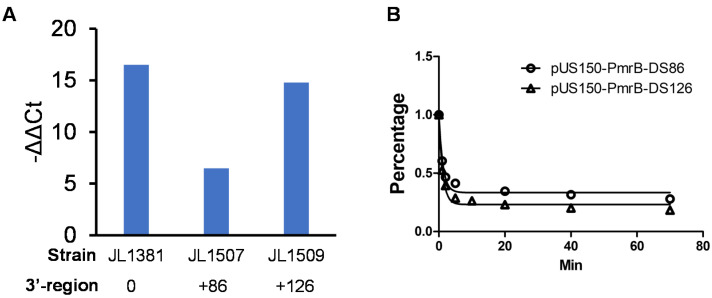
The effects of *pmrB* downstream region on the transcription and stability of *pmrB*. (**A**) The effect of *pmrB* downstream region on *pmrB* mRNA levels. The cloned *pmrB* of BL21 origin in JL1381, JL1507 and JL1509 includes 0, 86 and 126 bp downstream regions, respectively (indicated below strain name). Quantitative RT-PCR was performed to assess transcription level of *pmrB.* The JL1088 strain that carries parent plasmid pZE21 was used as a control. The *pmrB* mRNA levels in JL1381, JL1507 and JL1509 were compared with that in JL1088 (expressed as −ΔΔCt). (**B**) Decay curves of *pmrB* mRNA in JL1507 (TOP10/pUS150-PmrB-DS86) and JL1509 (TOP10/pUS150-PmrB-DS126) after transcriptional arrest by rifampicin. The mRNA level was measured by quantitative RT-PCR (detailed in *Materials and Methods*). The Y axis represents the ratio (percentage) of quantities of *pmrB* mRNA at each time point relative to that at the 0 min time point. Three independent experiments were performed. Data points are the mean value from measurements of triplicate cultures in one representative experiment, and fit with one-phase decay curves to calculate the mRNA half-life.

**Figure 3 microorganisms-09-00655-f003:**
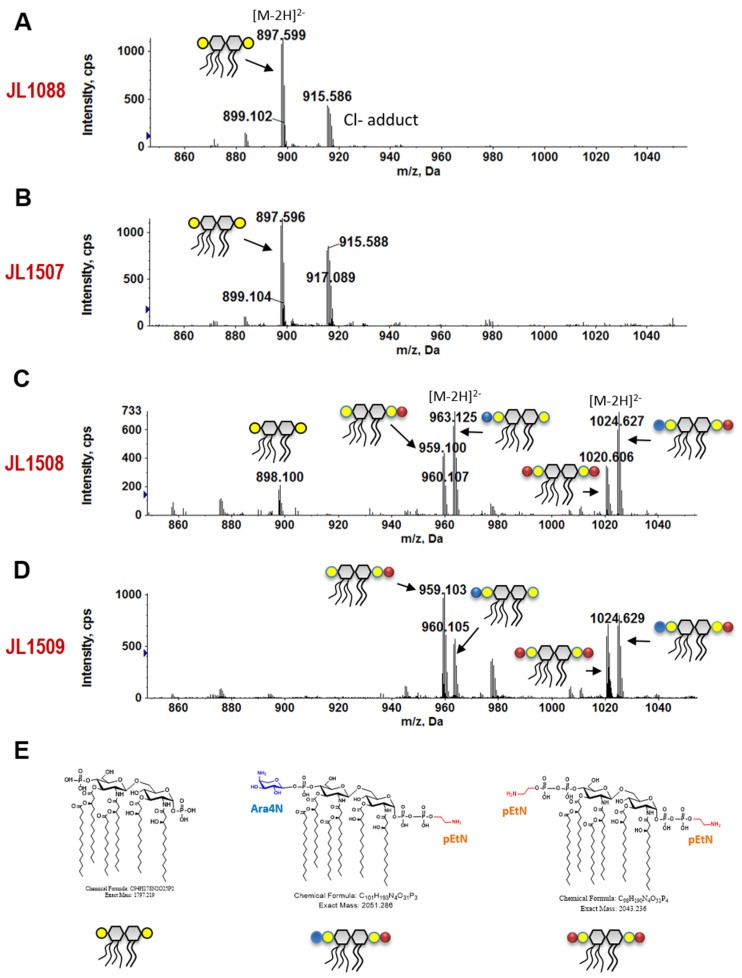
MS analysis of lipid A in the *E. coli* TOP10 strains bearing *pmrB*_BL21_ with different lengths of downstream regions. The doubly charged [M-2H]^2−^ ions of lipid A species are shown. (**A**) Unmodified lipid A in JL1088, the control strain containing pZE21 vector only; (**B**) Unmodified lipid A in JL1507, which contains plasmid pUS150-PmrB-DS86; (**C**) Extensive modifications of lipid A by Ara4N and pEtN in JL1508, which contains plasmid pUS150-PmrB-DS103; (**D**) Extensive modifications of lipid A by Ara4N and pEtN in JL1509, which contains plasmid pUS150-PmrB-DS126. The *pmrB* carried in three recombinant plasmids are diagramed in Figure 1A. (**E**) Representative chemical structures and the corresponding cartoons of hexa-acylated lipid A and its Ara4N and pEtN modifications.

**Table 1 microorganisms-09-00655-t001:** Key bacterial plasmids and strains used in this study.

Plasmids or Strains	Description	Source or Reference
Plasmids		
pZE21	Cloning and expression vector; kanamycin resistant (Kan^r^)	[22]
pUC19	Clone vector, ampicillin resistant (Amp^r^)	Invitrogen
pKD13	Template plasmid of kanamycin resistant cassette for gene disruption.	[23]
pSIM6	Ampicillin resistant. Heat inducible red recombinase expression plasmid, with a temperature-sensitive origin of replication	[24]
pZE21-EptA	pZE21 derivative containing *eptA* ORF	This study
pZE21-EptB	pZE21 derivative containing *eptB* ORF	This study
pZE21-CptA	pZE21 derivative containing *cptA* ORF	This study
pZE21-OpgEA	pZE21 derivative containing *opgE* ORF	This study
pZE21-PmrAB	pZE21 derivative containing *pmrA* and *pmrB* ORFs	This study
pZE21-PmrB	pZE21 derivative containing *pmrB* ORF	This study
pZE21-PmrD	pZE21 derivative containing *pmrD* ORF	This study
pPmrA-PmrB-ProP	pZE21 derivative containing PmrA(partial)-PmrB-ProP(partial)	This study
pUS150-PmrB-DS34	pZE21 derivative containing *pmrB* region from upstream 150 bp to downstream 34 bp	This study
pUS150-PmrB-DS86	pZE21 derivative containing *pmrB* region from upstream 150 bp to downstream 86 bp	This study
pUS150-PmrB-DS103	pZE21 derivative containing *pmrB* region from upstream 150 bp to downstream 103 bp	This study
pUS150-PmrB-DS126	pZE21 derivative containing *pmrB* region from upstream 150 bp to downstream 126 bp	This study
pUS150-PmrB-DS134	pZE21 derivative containing *pmrB* region from upstream 150 bp to downstream 134 bp	This study
pUS150-PmrB-DS176	pZE21 derivative containing *pmrB* region from upstream 150 bp to downstream 176 bp	This study
pUS150-PmrB_MG1655_-DS176	pZE21 derivative containing *pmrB*_MG1655_ region from upstream 150 bp to downstream 176 bp	This study
Strains		
BL21(DE3)	F^−^ *omp*T *hsd*S_B_ (r_B_^−^, m_B_^−^) *gal dcm* (DE3)	Stratagene/Novagen
TOP10	F^−^ *mcrA* Δ(*mrr-hsdRMS-mcrBC*) ϕ80*lacZ*ΔM15 Δ*lacX74 recA1 araD139* Δ(*ara leu*)*7697 galU galK rpsL* (Str^r^) *endA1 nupG*	Invitrogen
JL1374	TOP10/pZE21-EptA	This study
JL1375	TOP10/pZE21-EptB	This study
JL1376	TOP10/pZE21-CptA	This study
JL1377	TOP10/pZE21-OpgE	This study
JL1371	TOP10/pZE21-PmrAB	This study
JL1373	TOP10/pZE21-PmrD	This study
JL1365	Transformant #1 from functional cloning, Col^R^	This study
JL1366	Transformant #2 from functional cloning, Col^R^	This study
JL1367	Transformant #3 from functional cloning, Col^R^	This study
JL1368	Transformant #4 from functional cloning, Col^R^	This study
JL1369	Transformant #5 from functional cloning, Col^R^	This study
JL1397	TOP10/pPmrA-PmrB-ProP	This study
JL1088	TOP10/pZE21	This study
JL1381	TOP10/pPmrB, containing *pmrB* ORF only	This study
JL1431	TOP10/pUS150-PmrB-DS176	This study
JL1432	TOP10/pUS150-PmrB_MG1655_-DS176	This study
JL1611	TOP10/pUS150-PmrB-DS34	This study
JL1507	TOP10/pUS150-PmrB-DS86	This study
JL1508	TOP10/pUS150-PmrB-DS103	This study
JL1509	TOP10/pUS150-PmrB-DS126	This study
JL1444	TOP10/pUS150-PmrB-DS134	This study
JL1435	BL21(DE3), *phoQ*::*kan*	This study
JL1436	BL21(DE3), *pmrB*::*kan*	This study

**Table 2 microorganisms-09-00655-t002:** Major primers used in this study.

Primer	DNA Sequence (5′-3′) ^a^	Product Size (bp) ^b^	Target Gene/Region and Function
EptA_F	ATGTTGAAGCGCCTACTAAAAAGAC	1644	*eptA* ORF
EptA_R	CGCGGATCCTCATTCACTCACTCTCCT (*BamH*I)		
EptB_F	ATGAGATACATCAAATCGATTACAC	1692	*eptB* ORF
EptB_R	CGCGGATCCTTAGTTAGCCGCTGCCTC (*BamH*I)		
CptA_F	ATGCATTCCACAGAAGTCCAGGCT	1734	*cptA* ORF
CptA_R	CGCGGATCCTTACTGATTACCCACCTG (*BamH*I)		
OpgE_F	ATGAATTTAACCCTCAAAGAATCGC	1584	*opgE* ORF
OpgE_R	CGCGGATCCTTAAGGTTTCGGGTCG (*BamH*I)		
Prbas_F	AAATTCTGATTGTTGAAGACGATAC	1766	*pmrA*-*pmrB* ORF
PrEAbas_R	CGCGGATCCTTATATCTGGTTTGCCAC (*BamH*I)		
PmrB-F1	ATG CAT TTT CTG CGC CGA CCA ATA	1092	*pmrB* ORF
PmrB-R	ATATGGATCCTTATATCTGGTTTGCCACGT (*BamH*I)		
p5Up-KpnI-F3	ATATGGTACCGACGCTGAATATGGGTCGCC (*KpnI*)	2897	pmrA(partial)-pmrB-proP(partial)
p5Up-SalI-R3	ATATGTCGACAAGTTTTTTTCCCGGGGGCTGA (*Sal*I)		
US150-PmrB-SalI-F1	ATATGTCGACGACATCTATAACTGGGACAA (*Sal*I)		
DS34-PmrB-BamHI-R1	ATATGGATCCCCGTGTTCAGCGTGCTGGTG (*BamH*I)	1271	*pmrB* ORF, with 150 bp upstream and 34 bp downstream
DS86-PmrB-BamHI-R1	ATATGGATCCCAGGTTAACGGAGGAGAGTG (*BamH*I)	1328	*pmrB* ORF, with 150 bp upstream and 86 bp downstream
DS103-PmrB-BamHI-R1	ATATGGATCCACGCGCATACTCTCCTCCAG (*BamH*I)	1345	*pmrB* ORF, with 150 bp upstream and 103 bp downstream
DS126-PmrB-BamHI-R1	ATATGGATCCCTTAAGGTTCACTTAATCTC (*BamH*I)	1368	*pmrB* ORF, with 150 bp upstream and 126 bp downstream
DS134-PmrB-BamHI-R1	ATATGGATCCATTGAACTCTTAAGGTTCACT (*BamH*I)	1376	*pmrB* ORF, with 150 bp upstream and 134 bp downstream
DS176-PmrB-BamHI-R1	ATATGGATCCGCTGAAACGGATGGCCTGAT (*BamH*I)	1418	*pmrB* ORF, with 150 bp upstream and 176 bp downstream
pZE_F	GAATTCATTAAAGAGGAGAAAGGT	N/A	Forward sequencing primers for pZE21 derivatives
pZE_R	TTTCGTTTTATTTGATGCCTCTAG	N/A	Reverse sequencing primers for pZE21 derivatives
PmrB(BL21DE3)_pKD13_F3	GCTTTGGCTATATGCTGGTCGCGAATGAGGAAAACTAATTGAATCTGATGTGTAGGCTGGAGCTGCTTCG	1403	Site-directed mutation of *pmrB*
PmrB(BL21DE3)_pKD13_R3	TTCAGCGTGCTGGTGGTCAGCAGCTTTCTTTATATCTGGTTTGCCACGTAATTCCGGGGATCCGTCGACC		
PmrB_F	AATGAACCCTCGACCAACAC	1376	Detect site-directed mutation of *pmrB* with k1
PmrB_R	CGCTGTCTTATCAGGCCAAT		
PhoQ(BL21DE3)_pKD13_F3	GTGATTACCACCGTTCGCGGCCAGGGCTATCTGTTCGAATTGCGCTGATGTGTAGGCTGGAGCTGCTTCG	1403	Site-directed mutation of *phoQ*
PhoQ(BL21DE3)_pKD13_R3	TTAACGTAATGCGTGAAGTATGGACATATTTATTCATCTTTCGGCGTAGAATTCCGGGGATCCGTCGACC		
PhoQ_F	TAATGGCAAAGTGGTGAGCA	1772	Detect site-directed mutation of *phoQ* with K1
PhoQ_R	TTCTGCCAGTGACGTTCAAG		
K1	CAGTCATAGCCGAATAGCCT		Common primer for detecting site-directed mutation
RT-PmrB-BL21SNP-F3	CATTGCCATTCACAGCGCCACCCGCA	180	RT-PCR detection of *pmrB*_BL21(DE3)_
RT-PmrB-BL21SNP-R3	TGCGTTTTCGCCAGCAGTTCCAGATGCA		
16S-F	AAGTTAATACCTTTGCTCATTGAC	118	16S rRNA internal control for RT-PCR
16S-R	GCTTTACGCCCAGTAATTCC		

^a^ Restriction sites are underlined in the primer sequence, and the names are identified in parentheses. ^b^ The amplicon size using wild type genomic DNA of BL21(DE3) or pKD13 as templates.

**Table 3 microorganisms-09-00655-t003:** Functional cloning of colistin resistance determinants from *E. coli* BL21(DE3).

Strain	Insert Size (bp)	Genome Location in BL21	Annotated Genes and Organization ^a^
JL1365	10,660	210,471–208,308	*dnaE*
		3,818,298–3,821,491	*viaA-ravA-kup*
		1,144,385–1,147,209	*rne-yceQ*
		2,263,105–2,265,581	*arnF-pmrD-menE-menC*
JL1366	3926	4,242,936–4,239,011	*proP-**pmrB-pmrA**-EptA-adiC*
JL1367	6989	4,239,038–4,243,091	*proP-**pmrB-pmrA**-EptA-adiC*
		1,256,348–1,259,815	*kdsA-ldrA-ldrB-ldrC-chaA*
JL1368	3813	2,263,769–2,261,742	*arnT-arnE-arnF-pmrD*
		2,759,470–2,757,690	*syd-queF-ygdH*
JL1369	5269	1,879,745–1,877,377	*purT-eda-edd*
		4,237,732–4,240,628	*proP-**pmrB-pmrA***

^a^ The highlighted genes (red text) are those known to involve polymyxin resistance. The genes encoding two-component regulatory system PmrAB (red bold face) are the focus of this manuscript.

**Table 4 microorganisms-09-00655-t004:** Colistin MIC and lipid A modification of the *E. coli* TOP10 constructs carrying *pmrB*_BL21_ with different lengths of 3′-downstream region.

Strain	Colistin MIC (µg/mL)	Lipid A Modification	
		pEtN	Ara4N
TOP10/pZE21	0.5	- ^a^	-
TOP10/pPmrB	0.5	ND ^b^	ND
TOP10/pUS150-PmrB-DS34	0.5	ND	ND
TOP10/pUS150-PmrB-DS86	0.5	-	-
TOP10/pUS150-PmrB-DS103	4	+	+
TOP10/pUS150-PmrB-DS126	16	+	+
TOP10/pUS150-PmrB-DS134	16	ND	ND
TOP10/pUS150-PmrB-DS176	16	ND	ND
TOP10/pUS150-PmrB_MG1655_-DS176	0.5	ND	ND
BL21, wild type	16	+	+
JL1435 (BL21, *phoQ*::*kan*)	16	ND	ND
JL1436 (BL21, *pmrB*::*kan*)	0.5	ND	ND

^a^ -/+: there isn’t/is a lipid A modification in the corresponding strain. ^b^ ND: not determined.

## Data Availability

No new data were created or analyzed in this study. Data sharing is not applicable to this article.

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
