# Peer review of "Critical Role of 3′-Downstream Region of pmrB in Polymyxin Resistance in Escherichia coli BL21(DE3)"

_microorganisms, 2021, doi:10.3390/microorganisms9030655_

Round 1
Reviewer 1 Report
In this manuscript Xu et al. sought to determine the mechanism of colistin resistance which is observed in the model organism E. coli BL21 (DE3). The authors generated a DNA library from BL21 chromosome and by transforming it into TOP10 cells they identified that the pmrB gene confers resistance to colistin in E. coli BL21 (DE3). The authors demonstrate that a single nucleotide mutation G361A in pmrB results in colistin resistance and also for the first time demonstrated that the 3’-UTR of pmrB mRNA is required for this resistance phenotype. Through mass spectrometry analysis they show that the authors demonstrate that pmrB 3’-UTR is required for Lipid A modification which is known to a common mechanism of colistin resistance. This is a very well done study with appropriate controls and the results of the study support their conclusions and is bound to open new avenues of study.
A few very minor points:
- Line 42: Change ‘one of drugs’ to ‘one of the drugs’
- Line 330: The phrase ‘which only does not have’ is a bit confusing. Please remove ‘only’
- Figure 3: Label for panel E is missing.
- The real time primers used in the study were specific to the G361A mutation, hence the authors mention that pmrB was barely detectable in JL1088 (line 363). It would be very interesting to see if the native pmrB levels in TOP10 are higher or the same as in BL21 using a primer that is not specific to G361A mutation. If pmrB levels in TOP10 are higher or the same as in BL21 then it could suggest that there is functional interdependence between 3’-UTR and G361A mutation of pmrA mRNA of BL21.
Reviewer 2 Report
Title: Critical role of 3’-downstream region of pmrB in polymyxin resistance in Escherichia coli BL21 (DE3)
Authors: Fuzhou Xu, Atsushi Hinenoya, Ximin Zenga, Xing-Ping Li, Ziqiang Guan, and Jun Lin
Reference: microorganisms-1134532
Article type: Research
Reviewer Comments:
The manuscript microorganisms-113453, entitled “Critical role of 3’-downstream region of pmrB in polymyxin resistance in Escherichia coli BL21 (DE3)”, describes a novel regulatory mechanism of PmrB-mediated colistin resistance in E. coli.
General comments:
1. The English language can be improved in terms of writing style, spelling, and consistency, namely concerning:
a) English language (please see the Specific comments section)
b) Sentence length: usage of very long sentences, decreasing text readability.
2. Scientific style can be improved in terms of writing style, and assertiveness, namely concerning:
a) Usage of non-scientific language
l Line: please consider removing “kills”
l Lines 63/69: please consider replacing “cues” with “signals”
l Line 76: please consider replacing “vendors” with “suppliers”
l Line 363: please consider removing “barely”
l Line 385: please consider replacing “drastically” by “significant”
l Line 418: please consider replacing “appears to”
l Lines 419/420: please consider replacing “is relatively more”
b) Lack of assertiveness:
l Line 60: “can decrease”
l Line 63: “could respond”
l Line 65: “can interact”
l Line 347: “could confer”
l Line 387: “may modulate”
l Line 448: “may also dephosphorylate”
l Line 454: “ also can shift”
l Line 460: “could not confer”
l Line 462: “may contribute”
l Line 467: “may also play”
l Line 470: “ can interfere”
l Line 472: “ also can serve”
l Line 495: “ may behave”
c) Over descriptiveness in the Material and Methods Section
l If the methods were used according to manufacturer instructions and/or what was described in previous works, such a lengthy description is not required. The text should be simplified to improve its readability.
d) Over usage of personal remarks in the Results section
l Line 264: “Disappointedly”
l Line 265: “This frustrating preliminary study”
l Line 273: “It is not surprising that”
l Line 287: “It was unexpected”
l Line 291: “This interesting findings”
l Line 398: “As expected”
e) Over usage of extensive self-citation.
Specific comments:
Line 19: please consider removing “, a common model organism for research.”
Line 20: please consider replacing “significantly increased resistance to colistin, of which three contain pmrB” by “significantly increased colistin resistance, three of them containing pmrB”
Line 24: please consider replacing “Lipid A analysis by mass spectrometry revealed” with “Lipid A analysis, by mass spectrometry, revealed”
Lines 37: please consider replacing “polycationic peptide antibiotic with broad-spectrum activity” by “polycationic peptide antibiotics with broad-spectrum activity”
Line 38: please consider replacing “Escherichia coli, P. aeruginosa, Acinetobacter baumannii” by “Escherichia coli, Pseudomonas aeruginosa, Acinetobacter baumannii”
Line 44: please consider replacing “Gram-negative pathogens,” with “Gram-negative bacteria,”
Lines 46-54: confusing text. Please consider re-phrasing this paragraph.
Lines 56-57: please consider replacing “ lipopolysaccharide (LPS) modification” with “LPS modification”. The acronym LPS was previously described in line 47. There is no need to describe it once more.
Line 64: please consider replacing “mildly acidic pH,” with “moderate acidic pH,”
Line 91: please consider replacing “Complementation” with “Transformation”
Line 92-95: please consider replacing “Genes of eptA, eptB, cptA, opgE, pmrD and pmrAB two component system, which are able to contribute to colistin resistance in E. coli [6,25], as well as the different length of pmrB, were cloned into the cloning vector pZE21 [22] and introduced to colistin susceptible host strain TOP10.” by “Genes of eptA, eptB, cptA, opgE, pmrD, pmrAB and different length of pmrB, were cloned into the cloning vector pZE21 [22] and introduced to colistin susceptible host strain TOP10.”
Lines 111-112: please consider replacing “cloning as described in our previous publication [26]. Total bacterial genomic DNA of BL21 were extracted using” by “cloning as previously described [26]. Total bacterial genomic DNA of BL21 was extracted using”
Lines 115-116: please consider replacing “determined by running on 0.8% ethidium bromide containing agarose gel.” by “determined on agarose gels (with 0.8% ethidium bromide).”
Lines 116-118: please consider replacing “The gel slices corresponding to fragments sizes ranging from 1 kb to 4 kb were excised and the DNAs were extracted from the gel slice using QIAquick® Gel extraction kit (Qiagen).” by “Fragments with sizes ranging from 1 kb to 4 kb were excised and the DNA was extracted using QIAquick® Gel extraction kit (Qiagen).”
Lines 119-121: please consider replacing “End-repaired fragments were subsequently purified from the reaction mix by using QIAquick PCR purification kit (Qiagen) for ligation. Subsequently, the purified end-repaired fragments were ligated to EcoRV-digested” by “End-repaired fragments were purified from the reaction mix by using QIAquick PCR purification kit (Qiagen)and ligated to EcoRV-digested”
Line 143: please consider replacing “a generous gift from Dr. Donald Court” with “supplied by Dr. Donald Court”
Line 147: please consider removing “to remove methylated template plasmid”
Lines 151-152: please consider replacing “one microliter” by “1 µL”
Line 163: please consider replacing “as described previously” by “as previously described”
Lines 171-122: please consider replacing “as described previously” by “as previously described”
Line 178: please consider replacing “as described previously” by “as previously described”
Line 197: please consider replacing “as described previously” by “as previously described”
Lines 221-222: please consider replacing “described in our previous publications [34,35]” by “as previously described [34,35]”
Line 246: please consider replacing “The qRT-PCR data which were used to calculate the pmrB mRNA half-life was analyzed” by “The qRT-PCR data used to calculate the pmrB mRNA half-life were analyzed”
Line 250: please consider replacing “as described previously” by “as previously described”
Lines 254-255: please consider replacing “approach described in our recent Mcr-1 study [21]” by “approach previously described”
Lines 262-264: Not a Result, please consider moving this text to the Discussion Section.
Line 342: please consider removing “as indicated in”
Lines 352-353: Not a Result, please consider moving this text to the Discussion Section.
Line 365: please consider replacing “which contains pmrB ORF only,” by “which only contains pmrB ORF,”
Lines 385-388: Not a Result, please consider moving this text to the Discussion Section.
Lines 462: please consider replacing “To test this,” with “To test this hypothesis,”
Scientific comments:
Lines 352-359: If the results are preliminary, they should either be developed or not presented in a scientific manuscript.
Round 2
Reviewer 2 Report
After revision, the vast majority of the comments proposed by the reviewer were taken into consideration. After this process, the overall quality of the manuscript improved, being now ready for publication.